# Student and teacher performance during COVID-19 lockdown: An investigation of associated features and complex interactions using multiple data sources

**Sine Zambach** [1]☯*, **Jens Ulrik Hansen** [2]☯

**1** Department of Digitalization, Copenhagen Business School, Copenhagen, Denmark, **2** Department of People and Technology, Roskilde University, Roskilde, Denmark

☯ These authors contributed equally to this work.

* sz.digi@cbs.dk

**Data Availability Statement:** Because the data is not fully anonymized, we cannot disclose it publicly, however data can be made available to researchers who meet the criteria for access to

## Abstract

Due to the COVID-19 pandemic, testing what is required to support teachers and students while subject to forced online teaching and learning is relevant in terms of similar situations in the future. To understand the complex relationships of numerous factors with teaching during the lockdown, we used administrative data and survey data from a large Danish university. The analysis employed scores from student evaluations of teaching and the students' final grades during the first wave of the COVID-19 lockdown in the spring of 2020 as dependent targets in a linear regression model and a random forest model. This led to the identification of linear and non-linear relationships, as well as feature importance and interactions for the two targets. In particular, we found that many factors, such as the age of teachers and their time use, were associated with the scores in student evaluations of teaching and student grades, and that other features, including peer interaction among teachers and student gender, also exerted influence, especially on grades. Finally, we found that for non-linear features, in terms of the age of teachers and students, the average values led to the highest response values for scores in student evaluations of teaching and grades.

## 1. Introduction

COVID-19 caused disruption across the globe and had a devastating impact on many sectors, forcing policymakers around the world to impose partial or full lockdown. This included the education sector, which was forced to make a rapid transition to online education.

The change had a significant impact on both student and teacher conditions, but it is still unclear exactly how the lockdown affected student and teacher performance. In this paper, we aim to fill this research gap by examining the impact of the pandemic on both student and teacher performance.

More specifically, our overall research question is: What features are associated with student and teacher performance during the COVID lockdown in the spring of 2020?

confidential data under the GDPR. Additionally, we can create subsets of aggregated data if asked for. Data access requests may be sent to insight@cbs.dk.

**Funding:** The authors received no specific funding for this work.

**Competing interests:** The authors have declared that no competing interests exist.

Thus, our study seeks to provide a comprehensive analysis of the complex interactions between lockdown policies, online education, student demographics, teacher characteristics, and student and teacher performance. By examining both student and teacher performance, we aim to provide insights into the broader implications of pandemic-induced changes in teaching and learning, beyond the obvious fact that students' and teachers' daily lives changed along with their subjective experiences. The actual changes in student and teacher performance are critical to the quality of education and any future efforts to improve teaching and learning.

To answer our research question, we combine several different data sources (e.g., administrative data, survey data) to gain new insights into the following sub-questions:

- which linear, non-linear, and interacting features correlates with the performance of students measured by course grades?

- which linear, non-linear, and interacting features correlates with the performance of teachers measured by Students' Evaluation of Teaching (SET) scores?

Our data was collected at a Danish university, Copenhagen Business School (CBS), for the spring of 2020. Like all higher education institutions in Denmark, CBS officially switched to fully online teaching on March 16, 2020. Later, CBS announced that exams would also be held online. This primarily affected oral exams, which were mainly changed to written assignments at home, while written exams without access to books and notes on campus were mainly changed to assignments at home, where books were allowed.

In addition, teachers had to adapt their teaching to online formats such as webinars, prerecorded videos, online forums, quizzes, online collaborative documents, or other digital workarounds. While all teachers had access to a computer and the Internet at home, the quality varied, as did their IT skills and experience with working online. Students faced similar circumstances. Moreover, the teachers had to conduct online lessons in a potentially poor working environment in their own homes, while being influenced by the higher level of anxiety in society about the new virus, as well as the obstacles of homeschooling and caring for younger children at home.

As a result, we hypothesize that the lockdown and the shift to online education may have had different effects on student and teacher performance. Additionally, we expect that a variety of features, including student and teacher demographics and peer interaction, contributed to educational outcomes, although not all in a linear fashion.

To address this, and answer our research question, we build several linear regression and random forest models with the two targets: student grades and Student Evaluation of Teaching (SET) scores. We included numerous features measuring the conditions described above in an attempt to see which ones correlated with the targets. To further investigate non-linear and interaction effects, we used interpretable machine learning methods such as feature importance values, Friedman and Popescu's H-statistics, and various partial dependence plots.

## 2. Literature review

While the extent to which the COVID-19 pandemic affected the education sector is unprecedented, the need for schools to provide online instruction has a parallel in the 2003 SARS outbreak in China, Hong Kong, and Taiwan. These countries were forced to impose closures for several months and introduce online teaching. This transition to emergency online teaching has been studied by Fox [1] and McNaught [2], for example, who found that teachers had to spend more time teaching using information and communication technologies and less time on informal activities.

Seventeen years later, during the COVID-19 lockdown, the Internet, video-conferencing, and learning management systems were more common and widely available, although not everywhere in the world [3]. In addition, many university faculty had experience teaching online before the involuntary transition to online teaching and were able to share their experiences with colleagues around the world, for example in editorials, technical notes, and pre-prints at the onset of the pandemic [4–7]. Nevertheless, the everyday life of teachers was clearly affected by e.g. techno-stress, difficulties in meeting students' needs in a digital context, self-efficacy, lack of information and communication technology skills, self-efficacy, time needed to care for their own children, and gender differences [8–15]. These studies mainly use various qualitative methods to investigate the conditions of teachers during the COVID-19 lockdown, leaving any quantitative evidence of a potential change in student or teacher performance desired.

Although there seems to be a small gap in the quantitative research on changes in student and teacher performance, and what correlates with these changes during the COVID-19 lockdown, a few quantitative studies exist. Tailored student surveys were used to analyze student perceptions of teaching and found no or reduced SET scores [16–18]. In one study, characteristics such as female students, female teachers, and younger teachers were positively correlated with the SET scores [18]. On a course level, students were happy for the flexibility, but had great concerns regarding decreased in learning value, less attention and effort facilitated and academic quality [16, 17].

In addition, student performance during COVID-19 has been studied using quantitative methods by [19–23]. Some of these studies suggested that student demographics, such as gender and nationality, had no effect on changes in performance during COVID-19 at universities in the USA, the Netherlands, Spain, and Egypt [19–21, 23]. Others have suggested that there is a pattern of performance, particularly with respect to the socioeconomic background of students in an American university [22], which is consistent with earlier data, e.g. a Danish network study [24].

While these studies are important for understanding how the COVID-19 lockdown affected student and teacher performance, there is much more to be discovered. First, there is a need to increase validity and generality by examining conditions in other countries and universities using larger data sets. We remedy this by examining the effects of the COVID-19 lockdown on student and teacher performance at a Danish university, Copenhagen Business School, using comprehensive data from multiple sources on student performance, student demographics, courses, and teacher demographics, as well as an extensive survey of all teachers at the university (244 completed the survey) (see Section 3.1).

Moreover, the above mentioned studies only examined which features had a direct linear effect on student or teacher performance. Nevertheless, teaching and learning is a complex environment and the COVID-19 lockdown had diverse and subtle effects. Thus, it is hard to imagine that the effects on student or teacher performance were only linear and direct, leaving a gap in the research on student and teacher performance during the COVID-19 lockdown. We aim to fill this gap by using more advanced, but still interpretable, machine learning methods such as random forest together with feature importance values, Friedman and Popescu's H-statistics, and various partial dependence plots (see Section 3.2).

Machine learning methods have been applied in Educational Data Mining studies before, for example in [25–27]. However, these studies mainly showed that machine learning models outperformed linear models, implying that there must be non-linearity in the data. The studies did not go into detail about what the non-linear effects were and which features potentially interacted. Our study showcase a method for performing a detailed analysis of the particular non-linear effects and interactions.

## 3. Materials and methods

In this section we describe the data used (Section 3.1) as well as our statistical and machine learning methods (Section 3.2).

### 3.1 Data

In this study, we combined data from multiple sources concerning students, teachers, and courses at CBS, during the COVID-19 lockdown in the spring of 2020. We also used data on SET scores from the fall of 2018 onwards to calculate historical averages, and for grades; we used data dating back to 2010 to calculate historical grade point averages (GPAs). We only selected data from the various systems that was deemed relevant and pertained to teachers, students, or courses. Fig 1 provides an overview of the various types of data.

Student data is further described in Section 3.1.3 and course data in Section 3.1.4, while faculty data and teacher survey data are described in Section 3.1.2, and grade and SET data in Section 3.1.1. Finally, Section 3.1.5 describes the extensive data pre-processing we conduct and the final set of features.

For the survey data we obtained consent of the participants in connection with sending out the survey. All data was collected and stored in accordance with the Danish Data Protection Act. This means that all personal identifiers are pseudonymized in the data set from which this research takes departure. However, the first author did the pseudonymization and stored the keys on a separate server for which only she had access in accordance with the guidelines from the data protection officer of the first author's institution. As the study is non-invasive, we deem the only relevant ethical concern in regards to the human subjects that of privacy. Privacy in regards to research is governed by the Danish Data Protection Act (GDPR) administered by the Danish Data Protection Agency. Prior to the survey, we reported the fully study to the Danish Data Protection Agency and received approval under journal number 2015-57-0003. This included the approval of using the data that was not collected through the survey.

**3.1.1 Target data: Teacher and student performance.** We use student grades and SET scores as two different target features in our analysis, each one of which represents a study in itself with various models using features described in Section 3.1.5.

In the first analysis, which focuses on student grades, we infer associations with student performance during COVID-19 lockdown in the spring of 2020. We obtain grades for all courses with exams from 1 March to 31 August 2020. Grades are given based on the Danish grading scale, from lowest to highest: -3, 00, 02 (lowest passing grade), 4, 7, 10, and 12. The passing

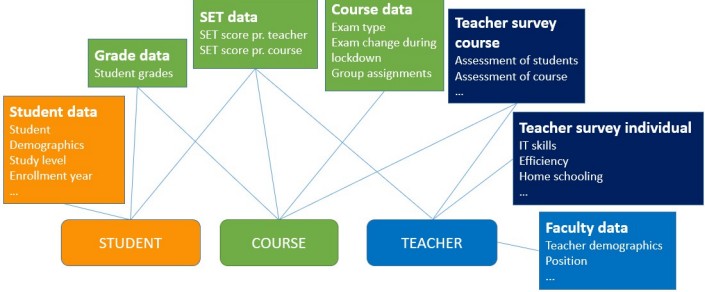

**Fig 1. Schematic view of the multilevel nested data.** The various colors indicate whether the data mainly concerns teachers, students, or courses, while the lines indicate the keys of the various datasets. The grades and student data are from the study administrative system, the SET data from the evaluation system, course data from the course catalogue, faculty data from the human resources system, and the two types of survey data are from a tailored survey.

grades correspond to grades in the European Credit Transfer System from E to A. In addition, some courses only assign the non-numeric grades of pass and fail. We remove all pass/fail data points and only include data points for students who received a pass since more students appear to have taken exams and submitted them blank in 2020 than in 2019, while more student failed in 2019 because of not showing up. As this may be related to the online format of exams, we exclude data points with failing grades. As the grading values are meant to be distributed normally (as confirmed by the histogram and quantile-quantile plot—see S1 and S2 Figs), we do not process the target further.

For the second analysis, we use the SET score on a five-point Likert scale as the target, which we treat as a numeric variable, even though it cannot be assumed to be normally distributed (as the histogram and quantile-quantile plot in S3 and S4 Figs show). The validity of our approach is supported by Derrick and White [28] and Norman [29] and appears to be the best method, since data is not uniformly distributed and therefore not suitable for a multinominal logistic regression.

Both grade and SET scores were in 2020 on comparable levels with 2019.

**3.1.2 Teacher survey data and teacher population.** The first author conducted the survey of CBS teachers in June 2020. Distributed using the survey software Qualtrics, the survey was designed to evaluate the first semester affected by COVID-19 to examine the challenges teachers faced and to identify any potential for future blended learning strategies. Most of the questions were based on a five-point Likert scale, though some provided answer categories for questions like "What tools do you plan to use next semester?". A few of the questions permitted free-text comments, but this study does not address those.

The survey questions were divided into two parts: personal questions for the teacher and then question about the specific courses they taught. The former contained questions about:

- The teacher's home workstation and surrounding environment, e.g., working position, stillness, and appearance or absence of children.

- The teacher's technical background, e.g., online teaching experience, IT skills, and attitude toward online teaching.

- Any didactic changes or adaptions, and tech use in general during the semester.

  The latter had questions about:

- The teacher's assessment of their own workload in each of their courses and whether their courses were successful.

- The teacher's assessment of, e.g., students and exams for each of their courses.

The survey was given to 837 teachers at CBS, including core and adjunct faculty who taught the spring semester of 2020. Of these, 328 completed the first part of the survey on teacher background information, while 244 completed the second part. Table 1 shows the teacher demographics of those who answered the second part compared to the total population of teachers who taught in the spring semester of 2020.

The population of teachers who completed the part of the survey on courses is significantly different (according to a Student's t-test) from the background population regarding age and number of courses taught.

The teachers (n = 244) who completed the part of the survey on courses represent all of the 11 departments at CBS and their response rate ranged from 21–42%, which indicates that each department is fairly represented. This is also the case for types of faculty, where the response rate was 17–50% for all faculty, with the lowest percentage among part-time faculty. The

**Table 1. Demographics of the teachers.**

| Faculty | N | Answer rate | Mean age | Mean #courses | Female rate | International rate |
|---|---|---|---|---|---|---|
| Part-time | 78 | 17% | 51.9 (45.8) | 1.5 (1.4) | 29% (28%) | 19% (15%) |
| Junior fac. | 23 | 31% | 34.2 (34.4) | 1.5 (1.4) | 74% (59%) | 65% (55%) |
| Assist. Prof. | 26 | 43% | 37.4 (37.5) | 1.7 (1.9) | 35% (42%) | 69% (68%) |
| Assoc. Prof. | 75 | 50% | 46.9 (46.9) | 2.3 (2.1) | 35% (30%) | 41% (44%) |
| Professor | 42 | 42% | 54.4 (53.0) | 2.1 (2.2) | 19% (17%) | 26% (39%) |

Demographics of the teachers who completed the part of the survey on courses with the full population listed in parentheses.

population comprised 33 nationalities, primarily Danish (n = 153), German (n = 29), and Swedish (n = 6). There were 90 teachers (37% of respondents) of a different nationality than Danish who responded (unreported (n = 1)). For comparison, 30% of all teachers who taught the spring semester of 2020 were not Danish, which we designate as international.

Figs 2 and 3 illustrate the distribution of answers to personal and the course questions, though with "Do not know" responses removed. S1 Table contains all questions.

Due to data pre-processing, as described in Section 3.1.5, we narrow the teacher population down further for each of our two model targets. The dataset for the grade target includes data from 162 unique teachers, while the dataset for the SET score target includes 114 unique teachers. The target populations resemble the general population with respect to the teachers' age and the number of courses they taught. For gender and nationality (whether teachers are Danish or not), there are no overall significant differences between the background population and the two target datasets. S2 and S3 Tables present a comparison of the teacher demographics between the two datasets and the population of all teachers who taught in the spring semester of 2020.

**3.1.3 Student data population.** As of May 2020, there were 14,137 students enrolled at CBS, 49% females, 25% internationals, and their average age was 24.6 years.

For student grade performance, we only included students who received a passing numerical grade on an ordinary exam. Once other data cleaning was performed (Section 3.1.5), 6,702 unique students remained in the grade dataset and had relatively similar demographics compared to the entire population of students enrolled in the spring semester of 2020 (S4 Table). However, the population of the grade dataset is significantly different (according to a Student's t-test) from the background population regarding nationality, age, and program level. Most master's students who enrolled in 2018 were in the midst of writing their thesis in the spring semester of 2020 and therefore did not have any course grades that semester. This was also the case for bachelor's students who enrolled in 2017, although most of them were taking courses while writing their bachelor thesis.

For SET scores, we have fewer data as only a fraction of students evaluated their teachers. Once other data cleaning was performed (Section 3.1.5), only 1,581 unique students remained in the SET score dataset (compared to 6,702 in the grade dataset). S5 Table presents their distribution relative to the background dataset.

Overall, the population for the SET score is significantly different from the background population regarding age, gender, and program level.

**3.1.4 Course data.** From the course catalogue we obtain information about whether exams contained a group assignment, i.e., whether the students were required to perform group work (written or oral). Moreover, we obtain information about changes to exam formats made during the lockdown, such as from oral to written, or from a written exam on campus to a written exam at home.

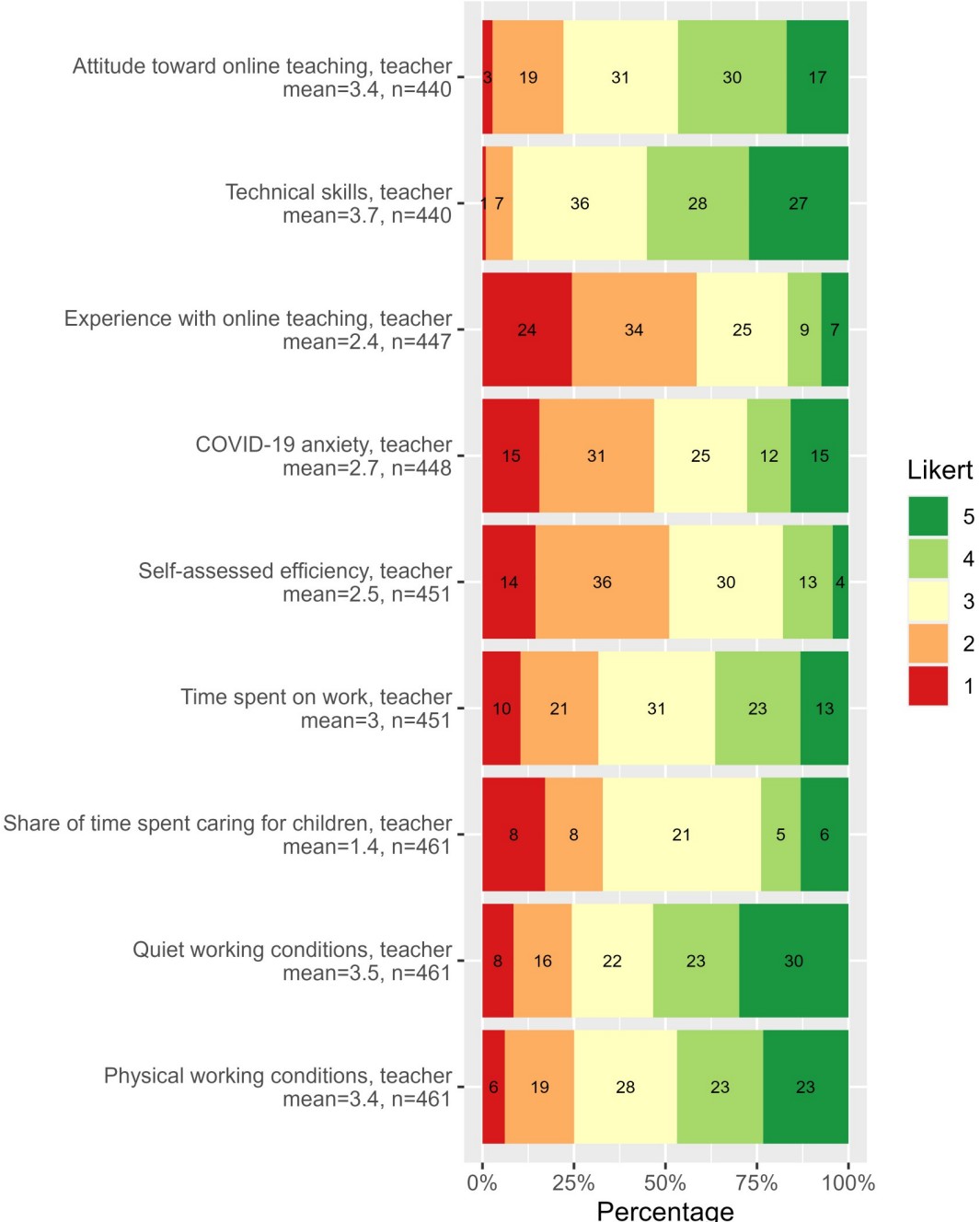

**Fig 2. Teachers' survey, personal questions.** Answers to background-related questions of the teacher survey. The questions with answers on the Likert scale are designed such that the higher value, the better. See also the full teacher survey description in S1 Table.

**3.1.5 Data pre-processing and final feature set.** Substantial data pre-processing is needed for the analysis and modeling performed in this paper. First, the data is transformed so that each data point concerns a student-teacher-course relationship. This is done to investigate the effect of teacher and course features on student grade performance and on SET scores.

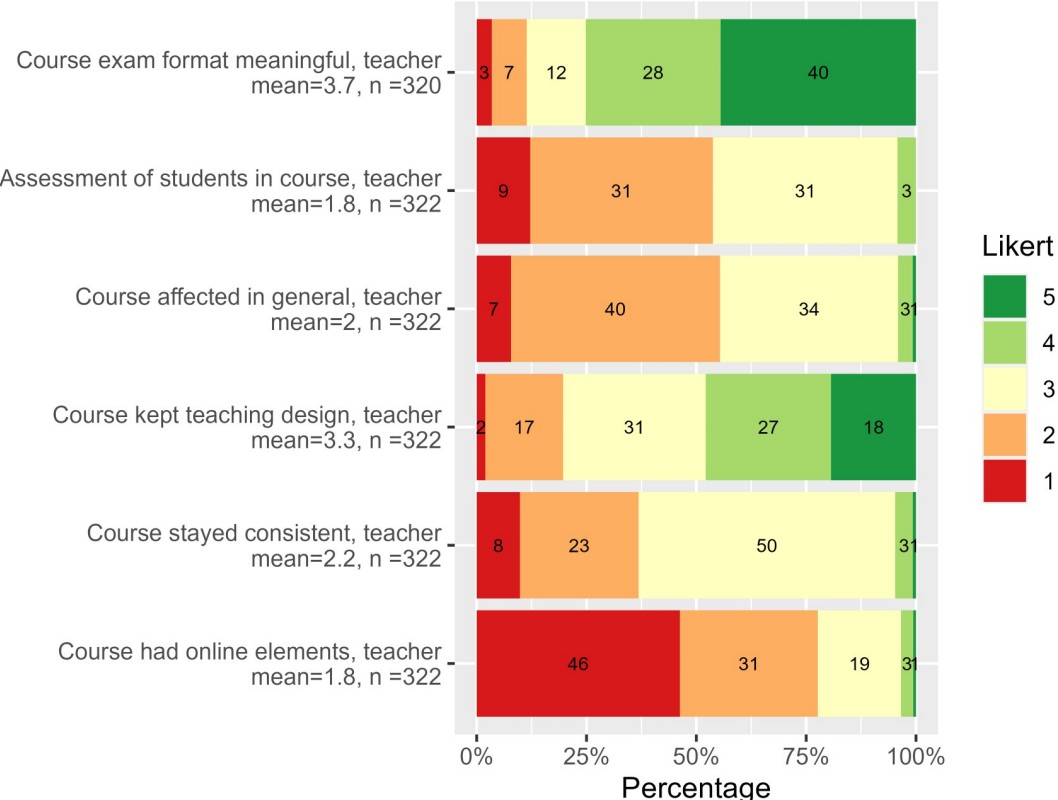

**Fig 3. Teachers' survey, course questions.** Answers to course-related questions of teacher survey. The questions with answers on the Likert scale are designed such that the higher value, the better. See also the full teacher survey description in S1 Table.

While data pre-processing is almost always needed, the fact that we used data from several systems require us to undertake numerous transformations, filtering, and joins. Moreover, most of the data is not collected purposefully for modeling in this paper, which is why transformation and feature engineering are necessary to create data of a format suited for our analyses. Finally, due to the aim of our analysis, part of the data was filtered out since we deemed it irrelevant to our analysis. S1 Text provides a complete list of the data transformation, feature engineering, and filtering of data.

The features of the analysis are based on both the teacher survey data and administrative data about teachers, students, and courses (see Fig 1). Table 2 lists the features in full.

First, there are administrative features concerning courses: required group work and changes in exam formats. Second, the administrative data also includes demographic measures such as gender, age, position (for teachers), department (for teachers), nationality, and program level (for students), some of which are mainly control variables. For instance, departments may have a variety of circumstances that are beyond the scope of our analysis. And, third, there is teacher assessments, which are explained thematically in Section 3.1.2.

We want to explore whether demographic measures or program level influenced student performance or whether it was a combination of both. Likewise, we want to examine teacher backgrounds more broadly, such as their technical skills, experience with online teaching, attitudes toward online teaching, and how these influenced student and teacher performance. We also use these features to potentially reveal various reasons for how a particular course was

**Table 2. List of features used for the models.**

| Administrative features | Teachers' assessment features |
|---|---|
| ***Students*** | ***Teachers' circumstances*** |
| Age, student | Children, teacher |
| Bachelor level, student | Share of time spent caring for children, teacher |
| Danish citizen, student | Physical working conditions, teacher |
| Female, student | Quiet working conditions, teacher |
| Enrollment year, student | COVID-19 anxiety, teacher |
| Historical GPA, student, control | ***Background*** |
| ***Teachers*** | Experience with online teaching, teacher |
| Age, teacher | Technical skills, teacher |
| Danish citizen, teacher | Attitude toward online teaching, teacher |
| Female, teacher | Time spend on work, teacher |
| Number of courses, teacher | Self-assessed efficiency, teacher |
| Job category, teacher | ***Didactic*** |
| Department, teacher, control | Institutional support, teacher |
| SET score before 2020, teacher, control | Peer consulting, teacher |
| ***Course*** | Experience with specific IT tools, teacher |
| Mandatory group work, course | Future intended use of specific IT tools, teacher |
| Exam from oral on campus to written | ***Course specific*** |
| at home, course | Course quarter, teacher |
| Exam from written campus to home, course | Course had online elements, teacher |
| | Course stayed consistent, teacher |
| SET score, course, control | Course kept teaching design, teacher |
| Previous year's GPA, course, control | Course affected in general, teacher |
| | Assessment of students in course, teacher |
| | Time use on course, teacher |
| | Course exam format meaningful, teacher |

List of features used for the models. Historical GPA and SET score averages are derived from course-specific data. Fig 1 provides an overview of the sources, their relationships, and a definition of color codes.

assessed since teachers may have used differing online mitigation strategies, depending on the course.

## 3.2 Statistical and machine learning methods

As previously mentioned, we build two datasets, one with student grades as the target and SET scores as the other. We built a variety of different linear regression and random forest models for each of the targets. We use a conservative significance level of $p < 0.0005$ for both linear regression models and variance importance measures because it corresponds approximately to a Bonferroni correction since we include 88 variables in our models.

**3.2.1 Linear regression models.** For both targets, we use the *lm* function in the statistical programming language R to fit the multiple linear regression models using the ordinary least square method [30]. Table 2 lists the features we used. We do not include any interactions between these features in the linear regression models. The resulting regression estimates are scaled by dividing by two standard deviations [31].

The scaled regression estimates, together with their estimation uncertainty, are visualized using dot-and-whisker plots [32], where the dots represent the regression estimate and the

whiskers span the 99.95% confidence interval. The significant regression estimates provides an idea of which features have a direct (linear) effect on our targets.

**3.2.2 Random forest regression models.**   Due to the complexity of the domain, it is unlikely that all effects are linear. Thus, to investigate the potential nonlinearity and interaction effects, we also train random forest models [33] on the same data. There are several advantages to using random forest models compared to other advanced machine learning models. First, they allow fairly interpretable output in the form of feature importance. Moreover, contrary to linear regression, random forest models can detect non-linear relationships and are less sensitive to confounders. Finally, they can detect interaction effects between several features.

We train the random forest models in R [30] using the ranger package [34], which is a rapid implementation of random forest models in accordance with Breiman [33]. We run the random forest model with 1000 trees and perform hyperparameter tuning of how many features to sample at each split (called *mtry* in ranger) and the minimum node size of the final trees. We use the *tidymodels* [35] framework to perform this hyperparameter tuning with a grid search and 10-fold cross validation.

We use the random forest models to extract feature importance values (representing direct effects) to compare with the multiple linear regression estimates. For feature importance, we use impurity importance from the ranger package since it measures importance as a decrease in impurity, where impurity is measured as the variance of the target [34]. That is, the high importance of a feature means that it has been used in numerous tree splits to create a high decrease in impurity, which intuitively signals the high importance of the feature for predicting the target. The significance of the feature importance is also calculated in the form of p-values using the ranger package (based on a permutation method originally suggested by [36]) and 6000 permutations.

To detect which features interact with other features to create effects on the targets in our random forest models, we use the R package *iml* [37], which provide multiple model agnostic tools for explainable artificial intelligence. More specifically, for our two random forest models, we use Friedman and Popescu's H-statistics [38] and partial dependence plots [39].

For each of the random forest models, we first calculate the total H-statistics for all features, which indicate whether and to what extent a feature interacts in the model with all the other features. Next, we select the features with the highest total H-statistics and calculate all two-way H-statistics for the selected features, which indicate any other features they interact with the most. Again, we select the seemingly most dominant two-way interactions and visualize them as partial dependence plots [40].

Since calculations of the total and two-way H-statistics are computationally heavy, sampling is used, which in turn introduce uncertainty. As a result, we run 10 to 25 different samples using cloud computing resources for each calculation of H-statistics to get a more robust result.

**3.2.3 Modeling strategy and evaluation.**   In contrast to classical statistical modeling, we adopt a hypothesis-free approach, as is common in machine learning and also adopted by others, such as [41]. In this approach we train predictive machine learning models without any prior hypothesis of what might have an effect, thus providing the possibility of discovering a larger variety of associations.

For both targets (student grades and SET scores), we first fit a multiple linear regression model with the entire set of features (Table 2) on the largest dataset without any missing values. Moreover, we also fit a random forest model for both targets, using the hyper-parameter tuning strategy explained above.

We introduce GPA prior to 2020 (*Historical GPAs, student*) in our feature set to control for it in the models with grade as target. Thus, any associations with grade, we find, are particular

to the spring semester of 2020. The same approach is used for the models with SET score as target, i.e. we introduce SET score of teachers prior to 2020 (*Historical SET score, teacher*). We also control for student assessments of the course in general (*SET score, course*) in the SET score models.

Apart from these four models (linear regression and random forest models for both grade and SET score targets), we built a collection of alternative models to validate the original choice of models. For student grade performance, we built alternative models using the following alternative input data:

- Excluding all features from the teacher survey

- Weighting of data points based on estimated teacher's share in the course

For the SET score models we also built alternative models using the following alternative feature sets:

- Excluding the overall course evaluation feature

- Including a historical, course mean evaluation

- Excluding all features from the teacher survey

In addition, we creat a random forest classification model instead of a regression model.

We train these alternative models to see if they show any substantial improvements in performance and to test the stability of the associations in the models. Table 3 shows the predictive performance of these models compared to the original models, while S2 Text describes the associations found in the alternative models in more detail.

We evaluate the predictive performance of both the linear regression and the random forest models using r-squared ($R^2$). We try to avoid overfitting in the random forest models by using out-of-bag samples upon which the reported $R^2$ is calculated [33]. The out-of-bag $R^2$ measure (OOB $R^2$) provides a realistic measure, allowing us to not reduce our training set by splitting it

**Table 3. Performance metrics for basic and alternative models.**

| Model | Type | Data | $R2$ | RMSE | OOB $R2$ | OOB RMSE |
|---|---|---|---|---|---|---|
| Grade | Linear regression | **Basic data** | **0.24** | **2.46** | | |
| | | w/o. survey data | 0.21 | 2.51 | | |
| | | w/ weights | 0.25 | 2.47 | | |
| | Random forest | **Basic data** | | | **0.32** | **2.33** |
| | | w/o survey data | | | 0.31 | 2.34 |
| | | w/ weights | | | 0.28 | 2.38 |
| SET score | Linear regression | **Basic data** | **0.52** | **0.71** | | |
| | | w/o course evaluation | 0.23 | 0.89 | | |
| | | w/ historical course mean | 0.52 | 0.71 | | |
| | | w/o survey data | 0.49 | 0.73 | | |
| | Random forest | **Basic data** | | | **0.48** | **0.74** |
| | | w/o course evaluation | | | 0.19 | 0.92 |
| | | w/ historical course mean | | | 0.42 | 0.78 |
| | | w/o survey data | | | 0.45 | 0.75 |
| | | Classification | | | - | - |

For the random forest models we used "out-of-bag" (OOB) samples to calculate $R^2$ and *RMSE*.

into testing and training sets, which would be undesirable due to our sample size. However, our sample size is large enough such that overfitting is not an issue in our linear regression models.

Additionally, we use both model types, along with the most prominent interactions, when evaluating the importance of each feature. Where the linear regression model outputs polarity of the estimate and significance of the linear dependencies, the random forest model exhibits the significance of the strength of the feature's influence through feature importance techniques on the responses/output grade and SET score, regardless of whether they are linear or not. Finally, interaction measures can point to features that might be affected heavily by interaction with other features, which also indicates the importance of a certain feature.

## 4. Results

In this section, we present the findings from our student grade and SET score models. The implications of the findings will be treated in the discussion. Table 3 shows the performance metrics for our basic and alternative models. We see no improvement in performance of the alternative models. Moreover, the changes in the associations were minor (S2 Text). Therefore, we will only report the findings from the basic models and limit the discussion mainly to these findings.

### 4.1 Student grade model results

The linear regression model performs worse than the random forest model with respect to both $R^2$ and the root mean square error (Table 3). This is unsurprising since the random forest model allows for non-linear and interaction effects, as discussed in Section 3.2.2.

Table 4 summarizes the regression estimates of the linear regression model, the random forest feature importance, and the random forest feature interactions. Fig 4 illustrates the scaled linear regression estimates (Section 3.2.1), Fig 5 illustrates the feature importance (Section 3.2.2) from the random forest model.

The only student features with a significant association with student grades in the linear regression model are age and enrollment year (controlled for historical GPAs at course and individual level). In the random forest model, we see that whether the student is a bachelor student, the enrollment year, and the student's gender have significant associations with grades. Furthermore, the student age feature receives a high feature importance, but the associated p-value turns out to be insignificant. Particularly *Age, student* (see also Fig 6) and gender (*Female, student* and *Female, teacher*) seem to interact with many other features, in addition to having a high level of feature importance (Fig 7). As another example of interaction effects (Fig 7), an exam change from written on campus to written at home (*Exam from written on campus to at home, course*) seems to have a negative effect for female students and a positive effect for male students.

At course level (administrative data), particularly two features have an effect, namely the change from *Exam from written on campus to at home, course* and from *Exam from oral on campus to written at home, course*. When an oral exam is changed to a written exam at home, the student's grade goes down in both the linear regression and random forest models. For *Exam from written on campus to at home, course*, the effect on the student's grade is negative in the linear regression model, while very slightly positive in the random forest model. However, in the random forest model, this last feature also interacts highly with several other features.

For the course-specific questions in the teacher survey, *Course exam format meaningful, teacher* has a negative effect in the linear regression and random forest models. The level of online elements before COVID-19 (*Course had online elements, teacher*) has no direct effect

**Table 4. Results for the grade models.**

| Feature | Linear Model Estimate | Feature Imp. (polarity) | Interaction (two-way H-statistics > 0.10) |
|---|---|---|---|
| Age, student | -0.32* | 11773 ↕ | Female, student (0.18)<br>Previous year's GPA, course (0.15),<br>Historical GPA, student(0.12) |
| Bachelor level, student | 0.07 | 241* ↓ | |
| Enrollment year, student | 0.25* | 1029*↕ | |
| Danish citizen, student | 0.13. | 541(↑) | Female, student (0.18) |
| **Female, student** | -0.03 | 1179* ↓ | Age, student (0.18),<br>Quiet working conditions, teacher(0.16),<br>**Exam from written on campus to at home, course (0.42)**,<br>Danish student(0.18) |
| **Number of courses taught, teacher** | 0.05 | 1029* ↑ | **Course had online elements, teacher(0.28)**, Assessment of students in course, teacher (0.15), Exam from written on campus to at home, course(0.14) |
| Female, teacher | 0.21. | 227*↑ | |
| **Quiet working conditions, teacher** | -0.42* | 824* ↓ | **Exam from written on campus to at home, course (0.27)**, Female, student (0.16), Will use other tools in future, teacher (0.11), Age, teacher (0.10) |
| Children, teacher | 0.70* | 170* ↑ | |
| Share of time spent caring for children, teacher | -0.78* | 500* ↕ | |
| Time spend on work, teacher | -0.09 | 565* ↕ | Course kept teaching design, teacher (0.14) |
| Technical skills, teacher | -0.35* | 925*↓ | |
| Attitude toward online teaching, teacher | -0.57* | 558* ↕ | |
| Assistant professor, teacher | 0.30* | 131* ↑ | |
| Part-time lecturer, teacher | 0.14 | 193* ↓ | |
| Professor, teacher | -0.41* | 246* ↓ | |
| Support from department, teacher | 0.63* | 386*(↓) | |
| Support from IT unit, teacher | 0.09 | 223* (↓) | |
| Support from study board, teacher | -0.29* | 135 (↑) | |
| Peer consulting, teacher | 0.59* | 389* ↑ | |
| Experience with own videos, teacher | 0.53* | 163* ↑ | |
| Experience with others' videos, teacher | -0.55* | 133 (↓) | |
| Experience with quizzes, teacher | 0.29* | 148*↓ | |
| Experience with shared documents, teacher | 0.37* | 211* ↑ | Female, student (0.11) |
| Experience with other tools, teacher | -0.09 | 237* ↓ | |
| Will not use digital tools in future, teacher | 0.69* | 64 ↓ | |
| Will use others' videos in future, teacher | 0.02 | 154 * ↓ | |
| Will use own video in future, teacher | 0.32* | 135 (↓) | |
| Will use quizzes in future, teacher | 0.04 | 175*↓ | |
| Will use forums in future, teacher | 0.03 | 191* ↑ | |
| Will use other tools in future, teacher | -0.37* | 365* ↓ | Quiet working conditions, teacher (0.11) |
| **Course had online elements, teacher** | 0.00 | 475 ↓ | **Number of courses, teacher (0.28)** |
| Course taught in Q3, teacher | 0.01 | 511* ↑ | |

*(Continued)*

**Table 4.** (Continued)

| Feature | Linear Model Estimate | Feature Imp. (polarity) | Interaction (two-way H-statistics > 0.10) |
|---|---|---|---|
| Course taught in Q4, teacher | -0.37* | 179* ↓ | |
| Course kept teaching design, teacher | 0.38* | 864*↑ | Course taught in Q3, teacher (0.19), Time spend on work, teacher (0.14), Course affected in general, teacher (0.13) |
| Course affected in general, teacher | -0.13. | 531*↓ | Course kept teaching design, teacher (0.13) |
| Assessment of students in course, teacher | 0.13. | 580*↕ | Number of courses, teacher (0.15) |
| Course exam format meaningful, teacher | -0.29* | 710*↕ | |
| **Exam from written on campus to at home, course** | -0.36* | 425*(↑) | **Female, student (0.42), Quiet working conditions, teacher(0.27)** |
| Exam from oral on campus to written at home, course | -0.87* | 301* ↓ | |
| Historical GPA, student | 2.07* | 24584* ↑ | Age, student(0.12), Previous year's GPA, course (0.13) |
| Previous year's GPA, course | 1.37* | 8701*↑ | Age, student (0.15), Historical GPA, student (0.14), Technical skills, teacher (0.12) |

Results for the grade models including both linear regression estimates, random forest feature importance and interactions. Features with both insufficient significance level in any of the models and no interaction are omitted. Significance level $p \leq 0.0005$ is annotated with * and $p \leq 0.05$ with. Interactions of two-way H-statistics >0.25 are in bold. The direction of association in the random forest model estimated from partial dependence plots are shown: ↑ positive effect, ↓ negative effect, ↕ mixed positive and negative effect.

in either the linear regression or random forest models; however it has a strong interaction with *Number of courses, teacher*. In this interaction, grade level is higher among those who taught many courses that had a small number of online elements (Fig 7). Finally, courses that kept their original design (*Course kept teaching design, teacher*) has a positive association with grade (in both the linear regression and random forest models), also interacting with *Course taught in Q3, teacher*, *Time spent on work, teacher*, and *Course affected in general, teacher*.

Among the teachers' background features, having children (*Children, teacher*) tends to have a positive effect on student grades in both the linear regression and random forest models, while *Share of time spent caring for children, teacher* has a negative effect (in both models for values above 40%). Additionally, being a female teacher has a very small positive effect on student grades in both models; however, it is only truly significant in the random forest model (*p*-value of 0.0035 in the linear regression models).

Having a high level of technical skills (*Technical skills, teacher*) and a positive attitude toward online teaching for a teacher *(Attitude toward online teaching, teacher)* seems to have a negative association with student grades in the linear regression and random forest models, while *Experience with own videos, teacher* and *Experience with shared documents, teacher* have a positive association with grades (in both models). Table 4 shows the remaining associations for experience and future use of tools and grades.

Finally, getting support from peers (*Peer consulting, teacher*) seems to have a positive association with grades. Both from the general question, *Have you given or received help from your colleagues?* and *Have you gotten help from your department?* has a highly positive association with the grades of the students in both models.

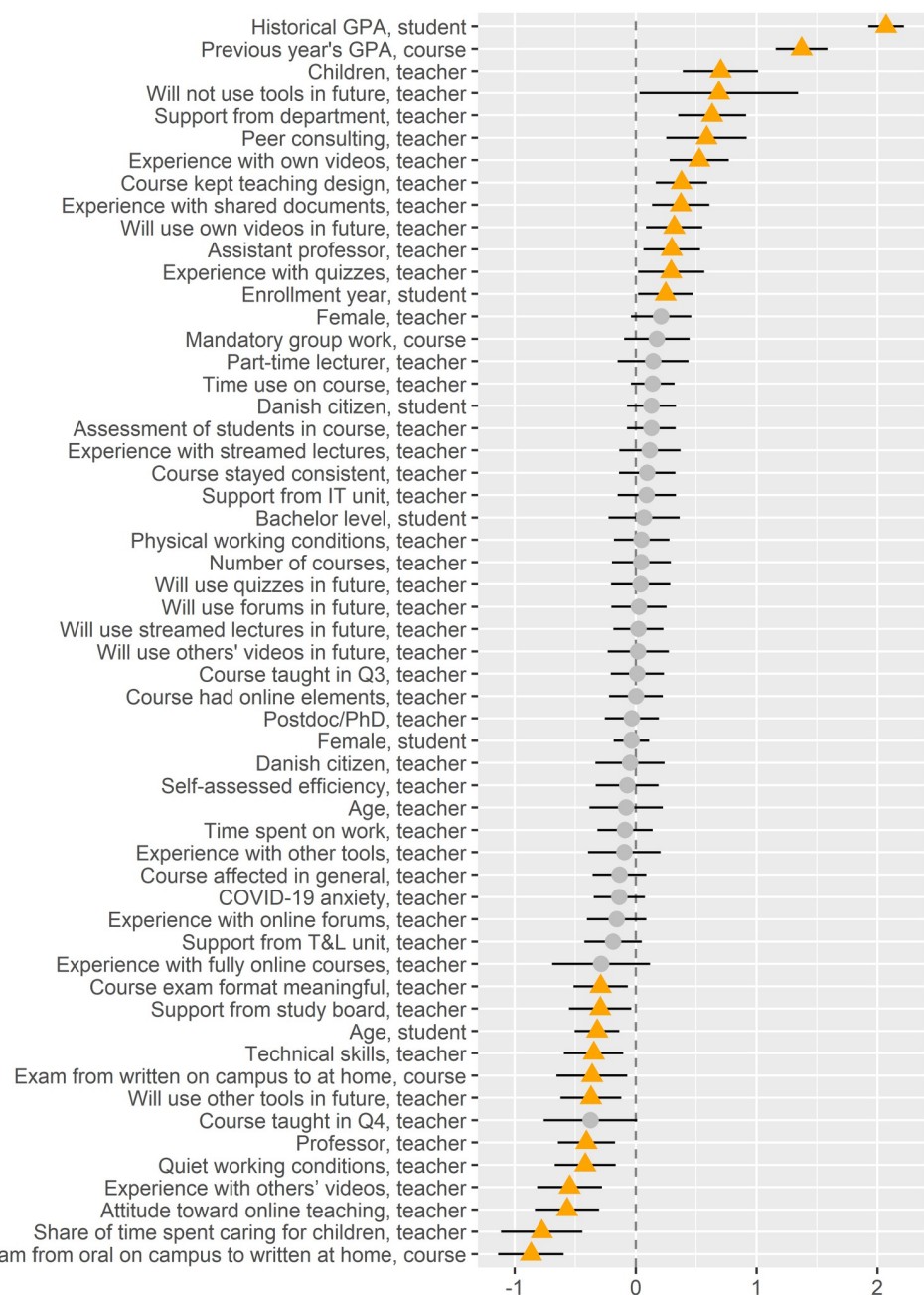

**Fig 4. Dot-and-whisker plot for the linear regression model with student grades as the target.** The dots represent the scaled regression estimates and the whiskers span the 99.95% confidence interval. The orange triangles represents the estimates that are significant at this confidence level.

## 4.2 SET score model results

Surprisingly, for the SET score models, the linear regression model performs better than the random forest model for $R^2$ and root mean square error (Table 3). Fig 8 illustrates the scaled linear regression estimates, while Fig 9 illustrates the feature importance from the random forest model.

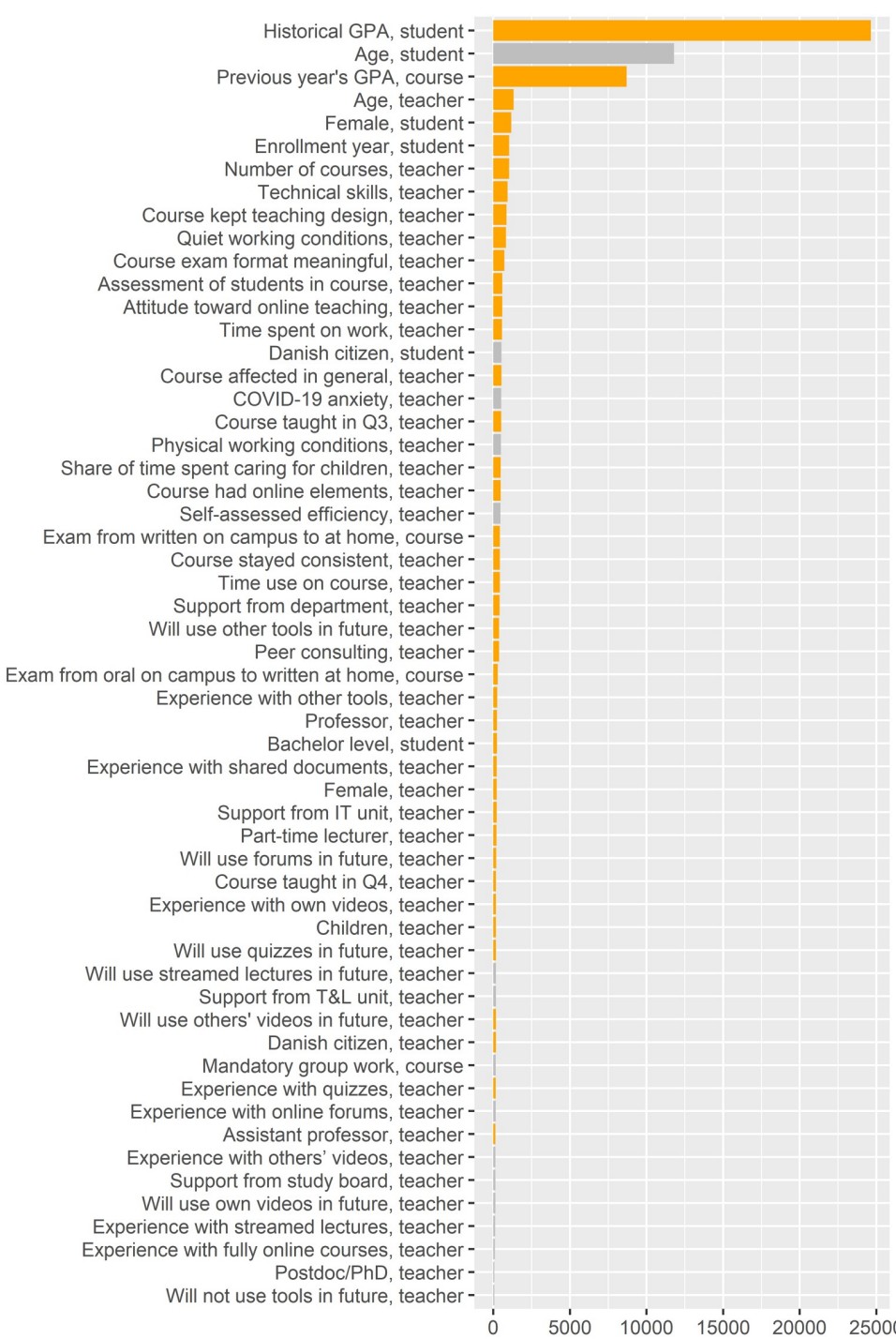

**Fig 5. Feature importance (impurity) plot for the random forest model with student grades as the target.** The orange bars show features that are significant at $p < 0.0005$ according to the method mentioned in Section 3.2.2.

Regarding the SET score model, we find one feature to be significant in both the random forest and linear regression models, namely teachers' overall assessment of students in the course (*Assessment of students in course, teacher*), which has a positive association with SET score. On the other hand, teachers who plan to use more videos from others (*Will use others'*

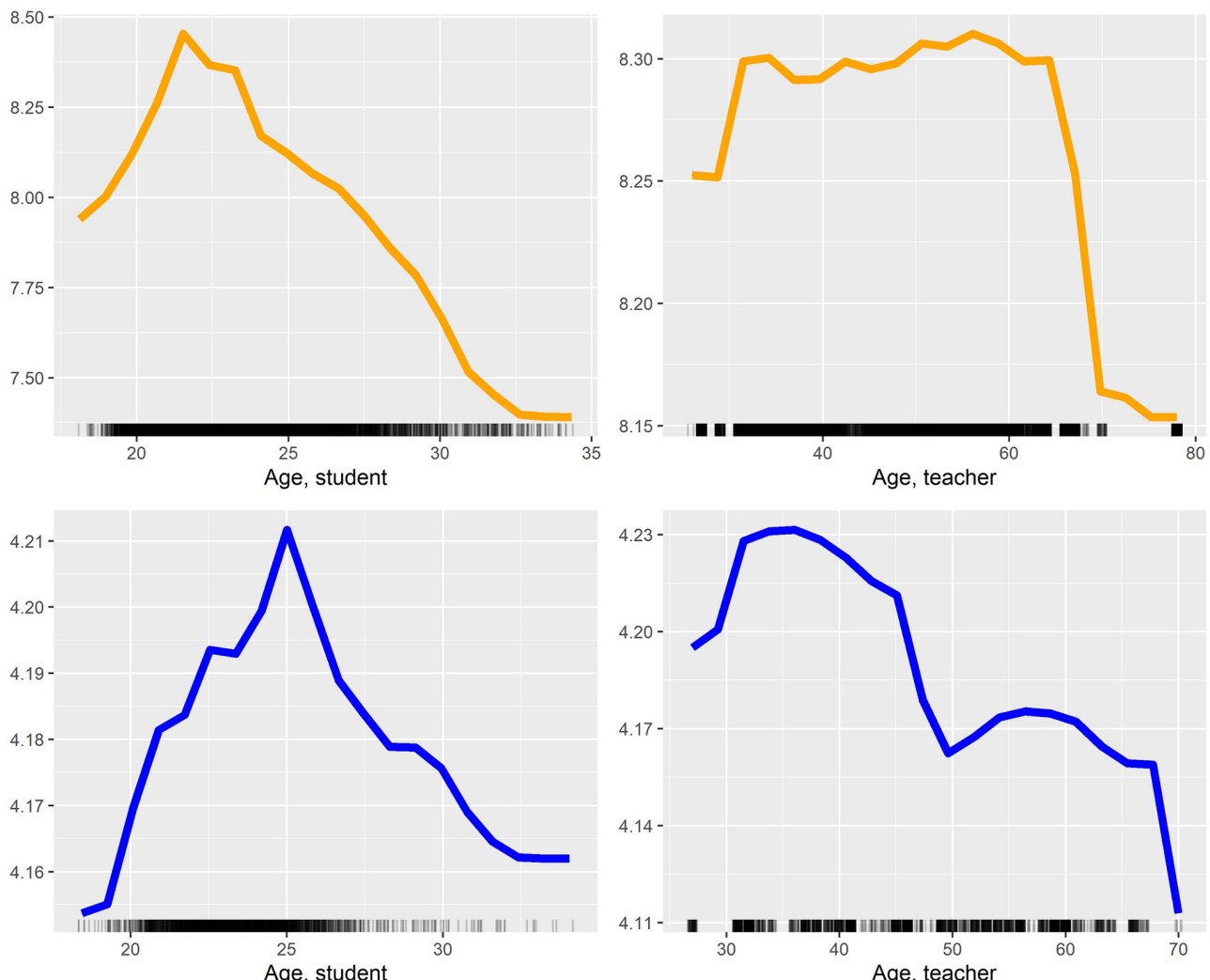

**Fig 6. Selected One-way partial dependency plots for age in both grade (A and C) and SET score model (B and D).**

*videos in future, teacher*) and being a Danish citizen (*Danish citizen, teacher*) have negative associations with SET score in the linear regression model but are insignificant in the random forest model. Finally, *Time use on course, teacher* is significant and positive in the random forest model but not in the linear regression model.

Besides this, several interactions are associated with the SET score in the random forest model (Table 5 and Fig 10). The *Age, teacher* (not significant in either models, however with a p-value of 0.012 in the linear regression model) have interactions with *Time use on course, teacher*, and *Self-assessed efficiency, teacher*, though the degree of these interactions are not high. Fig 10 indicates that a complex relationship exists between the *Age, teacher* and *Time use on course, teacher*, where a higher amount of time spent on the course tends to have a positive association with the SET score, particularly among teachers 30 to 48 years of age. Fig 6 shows the general link of age with SET scores.

Only one feature is significant in both random forest models for SET scores and student grades, namely *Assessment of students in course, teacher*, which have a positive effect. It is not

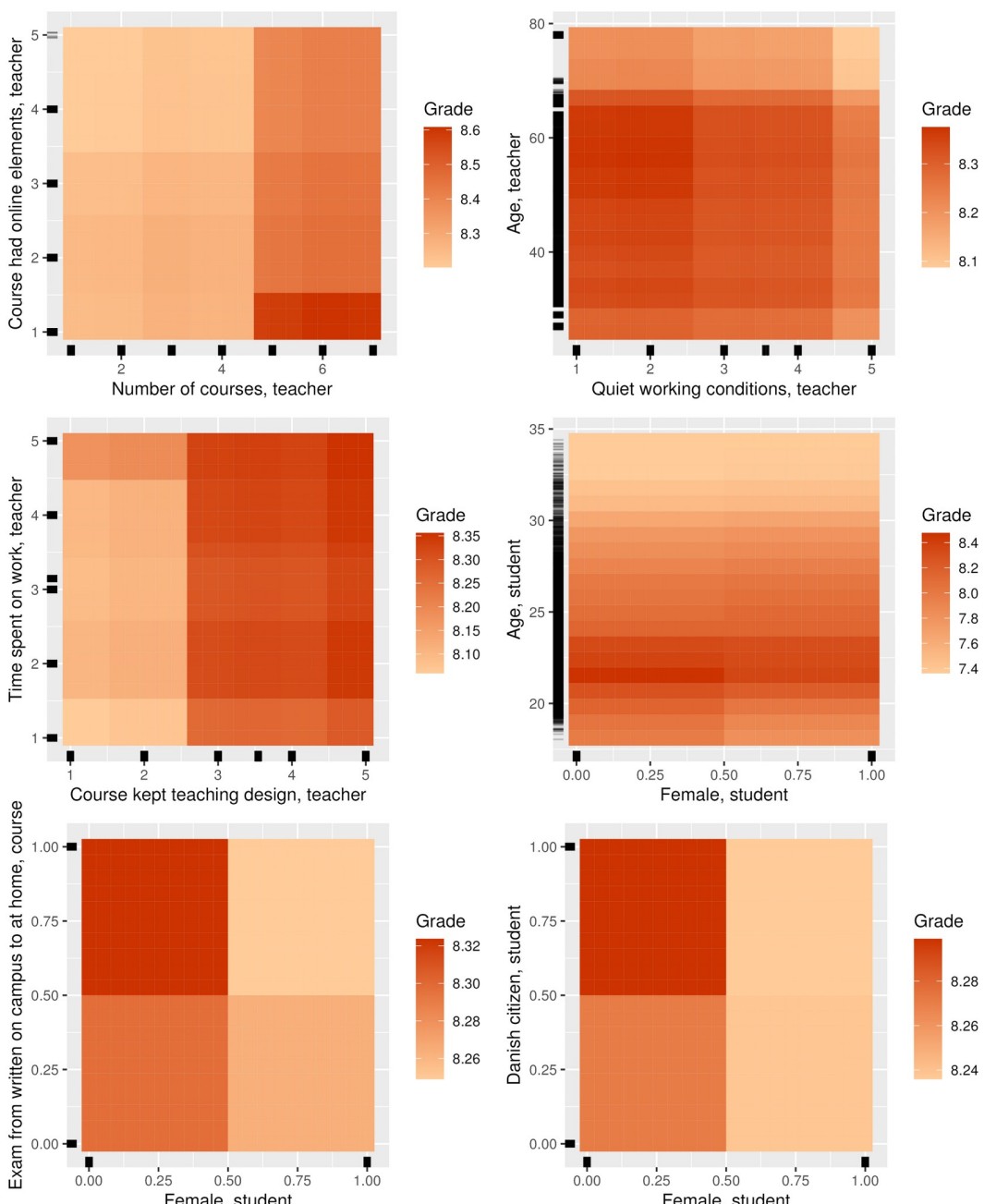

**Fig 7. Selected partial dependence plots for the random forest model with grade as target containing the features with the most interaction, and those that are clearly not linear.**

significant, however, in the linear regression model for student grades. Besides this, interactions for nonlinear features such as *Age, student*, *Number of courses, teacher*, and *Quiet working conditions, teacher* are found for both targets. However, these features interact with other features for both targets. Thus, there are almost no features, which are associated with both targets.

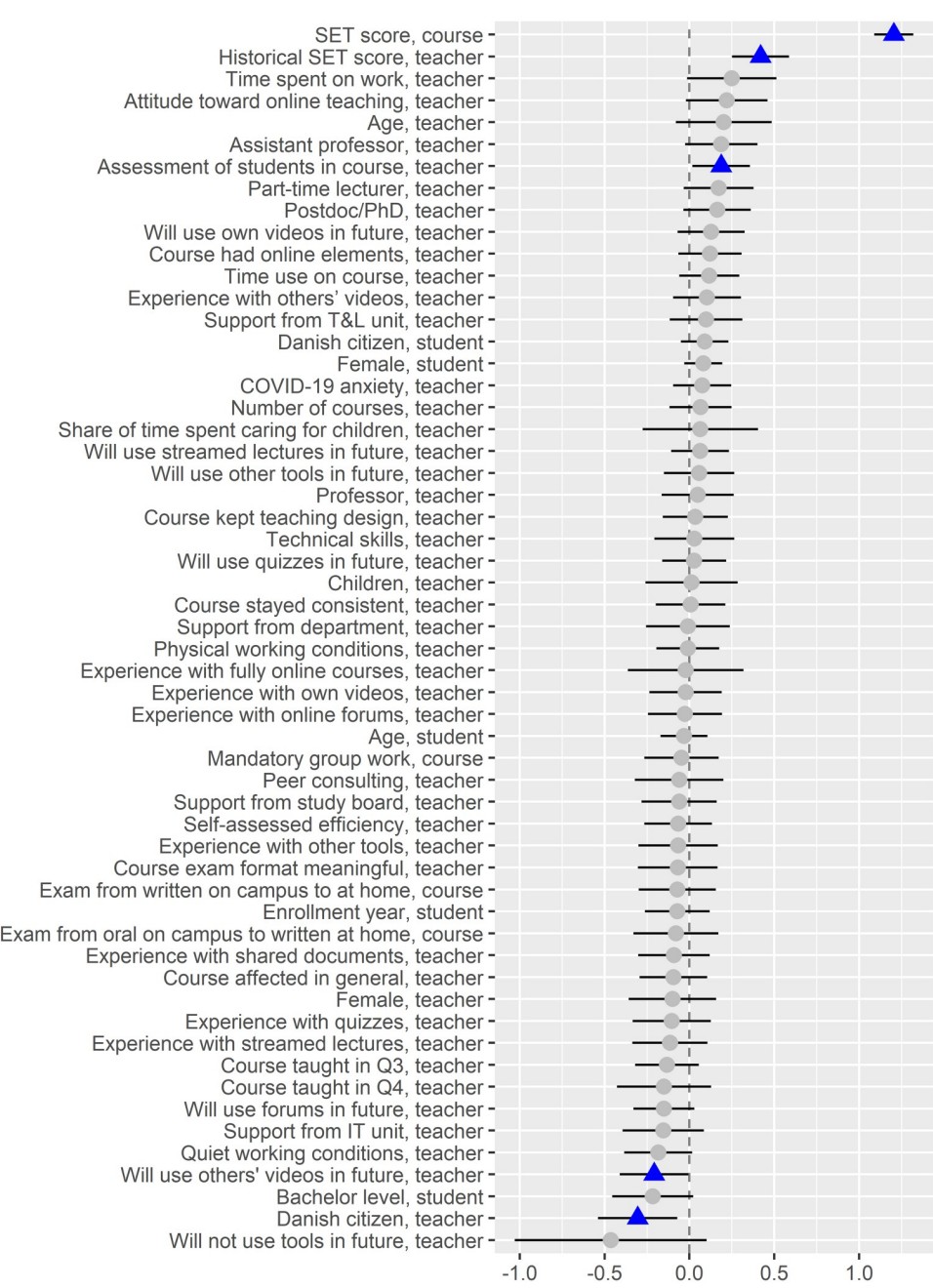

**Fig 8. Dot-and-whisker plot for the linear regression model with SET scores as the target.** The dots represent the scaled regression estimates and the whiskers span the 99.95% confidence interval. The blue triangles represents the estimates that are significant at this confidence level.

## 5. Discussion

In this section, after discussing our findings, we will address some general considerations for educational data mining and learning analytics before discussing the limitations of the study and mentioning suggestions for future research.

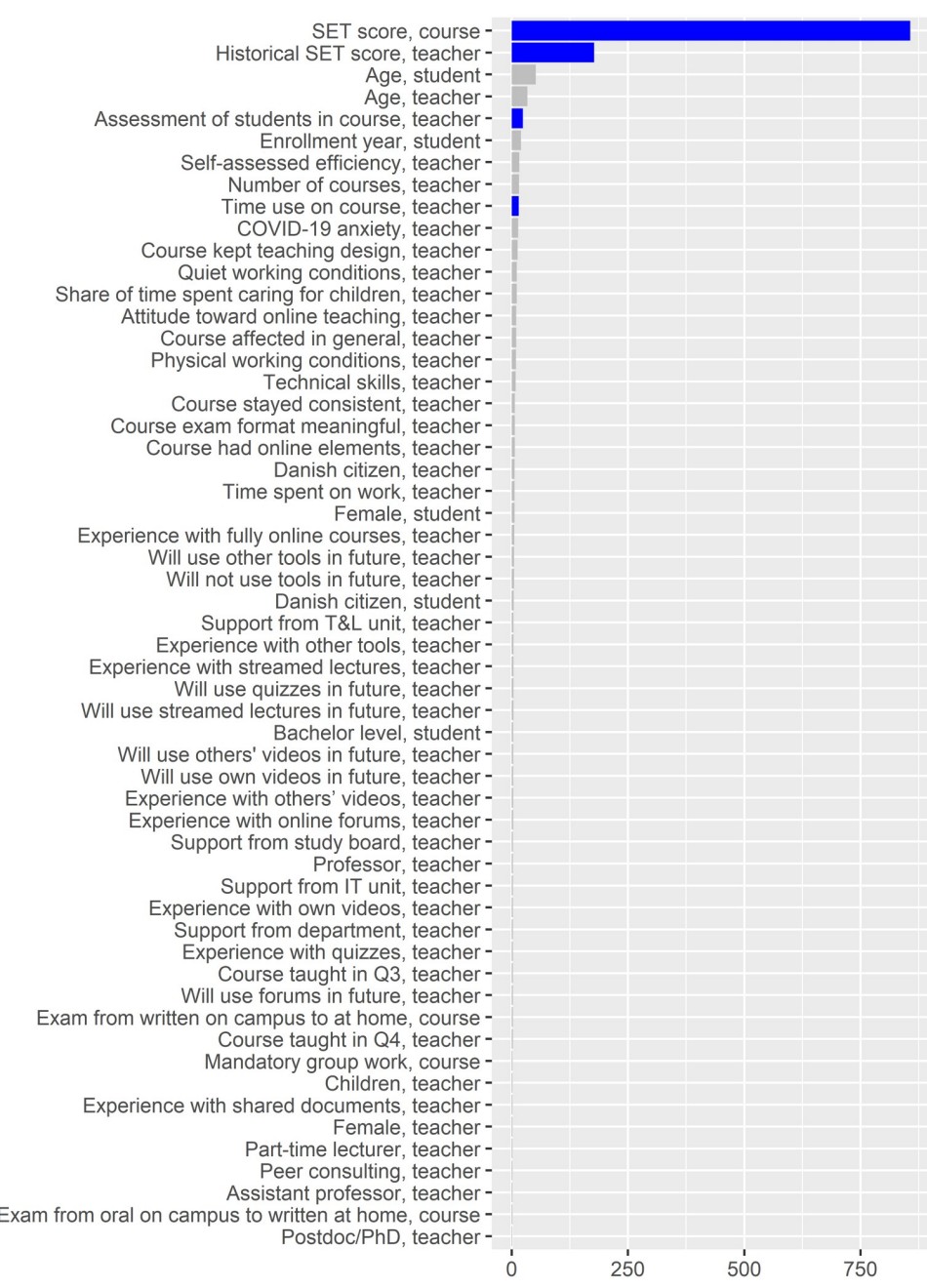

**Fig 9. Feature importance (impurity) plot for the random forest model with SET scores as the target.** The blue bars show features that are significant at $p < 0.0005$ according to the method mentioned in Section 3.2.2.

## 5.1 Findings

We start with the general assumption that a combination of teacher evaluations and student grades could be used to measure the quality of teaching in the belief that this approach will allow us to determine what influenced the quality of teaching during the COVID-19 lockdown, while simultaneously controlling for historical, personal, and course features.

Controlling for historical GPAs in the grade model and historical evaluations in the SET score model allows us to balance out biases that persisted to the same extent in 2020 as in 2019.

**Table 5. Results for the SET score models.**

| Feature | Linear Model Estimate | Feature Importance | Interaction (two-way H-statistics > 0.10) |
|---|---|---|---|
| Age, teacher | 0.20. | 33.9↕ | Time usage on course, teacher (0.11), Self-assessed efficiency, teacher (0.11)) |
| Assessment of students in course, teacher | 0.19* | 24.1*↑ | Quiet working conditions, teacher (0.12) |
| Time use on course, teacher | 0.12 | 14.6*↑ | Age, teacher (0.11) |
| Number of courses, teacher | 0.07 | 14.6↕ | Age, student (0.175) |
| Age, student | -0.03 | 50.8↕ | Number of courses, teacher (0.175) |
| Self-assessed efficiency, teacher | -0.07 | 15.5↕ | Age, teacher (0.11) |
| Quiet working conditions, teacher | -0.18. | 11.0↓ | Assessment of students in course, teacher (0.12) |
| Will use others' videos in future, teacher | -0.21* | 2.8↓ | |
| Danish citizen, teacher | -0.31* | 5.7↓ | |
| Historical SET score, teacher | 0.42* | 176.9*↑ | |
| SET score, course | 1.20* | 857.8*↑ | |

Results for the SET score models including both linear regression estimates, random forest feature importance and interactions. Features with both insufficient significance level in any of the models and no interaction are omitted. Significance level $p \leq 0.0005$ is annotated with * and $p \leq 0.05$ with. Interactions of two-way H-statistics >0.25 are in bold. The direction of association in the random forest model estimated from partial dependence plots are shown: ↑ positive effect, ↓ negative effect, ↕ mixed positive and negative effect.

For example, if there was an age bias in 2019 and it did not change in 2020, our model will show no association with the target and age. So, the correlation with age and the target shown in our model is resulting from the 2020 lockdown.

**5.1.1 Teachers' time use.**   The first theme we discuss in our findings concerns the preferences, skills, and working style of the teachers. Our findings show that ambiguity exists in the relationship between time use and the student grades, in the sense that the time teachers spend on course is not significant in either the linear regression or random forest models. The time teachers generally spent on work has a significant association with the grade in the random forest model, but the direction is neither clearly positive or negative, and the correlation in the linear regression model is not significant. However, there is a clear negative association between sharing responsibility for caring for children at home and grades in the linear regression model, and for those who had more than 40% of the responsibility, our findings show a clear negative association in the random forest model. While the teachers who have to take a larger share of caring for their children have a tendency to give lower grades, teachers who have children actually tend to give higher grades in general. This finding can be interpreted in multiple ways, though. Maybe teachers with children were not the ones who had the hardest time delivering quality teaching or maybe they had a harder time and were more forgiving toward their students. The fact that teachers without children also face challenges might also be supported by the fact that quiet working conditions are negatively associated with student grades.

For the SET score, we see no association with the amount of time spend on work in general, which is also the case for share of responsibility for taking care of children at home. However, we find a significantly positive relationship between whether teachers spent more time on a course than usual and the SET score they receive.

This suggests that the many teachers who spend additional time on their courses actually make a difference in terms of the students perception of their courses. The fact that this does not have a significant correlation with the student grades in our model can be due to other reasons, such as the teachers who spend more time on courses also expecting more of their

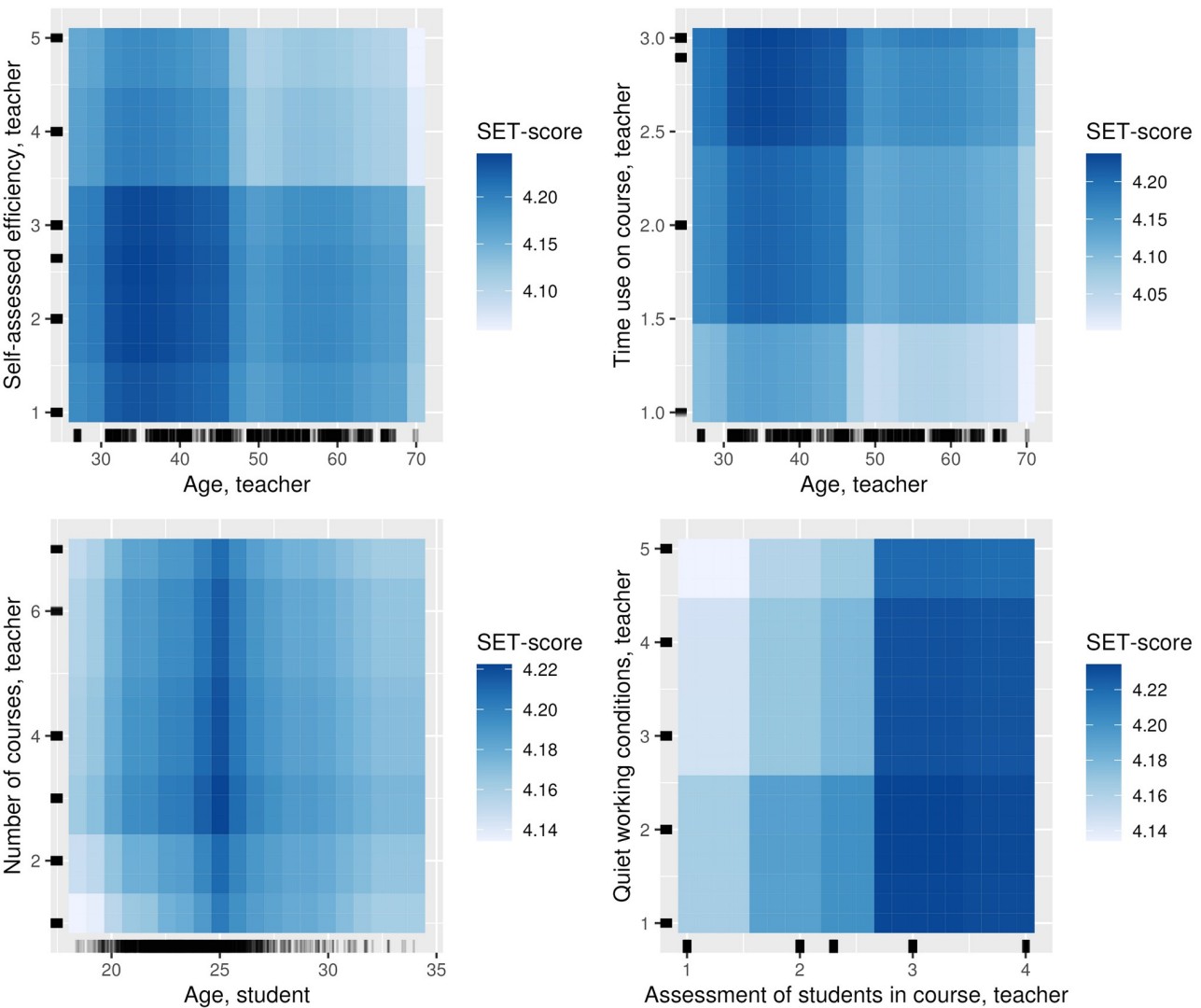

**Fig 10. Selected partial dependence plots for the random forest model with SET score as target containing the features with the most interaction, and those that are clearly not linear.**

students. In general, we expect the relationship between the time teachers spend on preparing courses and students performance in terms of grades to be complex, especially during a pandemic lockdown, and worth further examination in general. It is reassuring that the teachers who put in an extra effort regarding teaching are also rewarded in their SET score; however, it seems partially irrelevant if student learning is not also positively affected. Finally, it worth noting that approximately 80% of the teachers answered that they spent more time on courses than usual, which potentially could introduce a spurious effect.

This time balance should be further studied in relation to the research output of the same teachers since academic staff are required to both teach and do research. Do the extra time spent on adapting teaching to online teaching have a negative impact on their research? We expect so, but this is beyond the scope of this paper and a topic for future research.

**5.1.2 Teacher peer consulting.** Peer consultation among teachers is positively associated with grades in both the linear regression and random forest models, which may indicate that a

collective effort supports performance and is important to promote in times of isolation, both among peer teachers and students, to improve learning. It is widely accepted in the literature that peer collaboration is important for students [41–44]; however, we do not know much about the effect of peer consulting among teachers. We suggest that peer collaboration during crises appears to be equally, if not more, important for teachers, both in terms of the performance of their students and their own well-being. While peer consulting among teachers has a positive association with the student grades, there is no significant association with the SET score, which is a bit surprising, as consulting among teachers is expected to lead to better teaching, also from the students' point of view.

**5.1.3 Technical skills, dedication, and quiet working conditions.** One somewhat counterintuitive finding is that possessing high-tech skills, a positive attitude towards online teaching, and quiet working conditions for teachers seem to have a negative association with grades, but it must be remembered that grades are not a perfect measure of performance since they are also highly teacher dependent. For example, if teachers are struggling, they may feel sorry for their students and award higher grades.

When looking at the use of tools, there is a clear correlation between experience with a certain technology/didactics and the willingness to use it in the future. Our findings show that using one's own videos and the future intention to use one's own videos in teaching are positively associated with student grades, but that the shortcut of using others' videos has a negative association with grades. In addition, using other people's videos, for example from YouTube, even has a negative association with the SET score. Thus, this shortcut is questionable as a teaching tool, even though video recordings seem to have great learning potential. A first glance at the student comments in the teaching survey data shows that students are generally satisfied with their instructors' videos, which supports this analysis.

**5.1.4 Gender and grade.** One interesting finding relates to the debate about gender and grades. Female teachers tended to give slightly higher grades than usual—an association which is significant in the random forest model (and the alternative linear regression model using weights, as is described S2 Text), though not quite significant in the linear regression model (p = 0.0035), with a significance level of 0.0005.

Female students seem to receive lower grades than male students during the COVID-19 lockdown, especially among Danish students, which is confirmed by the random forest model but not the linear model. In addition, the gender association with grades seems to be larger when exams are changed from being on campus to taking place at home. The change seems to favor the younger men (Table 4 and Fig 7). A go-to explanation at first sight would be to couple this observation with the fact that it is easier for students to cheat during a home exam than at the university, which other studies have argued [45, 46]. These studies also find that the association is gender biased, in the sense that men cheat more often. This could potentially have an effect on student grades in our model. However, the fact that our analyses showed a negative linear association with student grades when written exams were moved from on campus to at home may indicate that cheating did not occur (successfully) very often.

Previous studies have shown that, under normal conditions, the skills of females are underrated [47, 48]. The present study showed that the gender bias was amplified during the COVID-19 lockdown in terms of grades. On the other hand, we found no significant change in the SET score of female teachers compared to male teachers during the lockdown.

**5.1.5 Do students at bachelor level perform better?** Bachelor students received a very slightly higher grade, as did students enrolled after 2018. Although newer students are more vulnerable to, for example, sudden changes in teaching because they are less used to study at an institution of higher education, they either managed to overcome the challenges satisfactorily or the teachers may have felt sorry for them, as suggested above.

In terms of students' age, we find that their grades peak around the age of 22, with lower grades shown for both younger and older students on average. However, it is likely not age, as such, that causes the differences in performance, but the fact that age often is a proxy for other aspects. At age 22, students may have reached a certain level of maturity but have not taken a long sabbatical or changed degree programs. On the other hand, more complex dynamics may also explain this. For instance, we see that student age is one of the features that interacts most with other features (Table 4). Among younger students, men seem to perform better than women (Fig 7).

Student age also have a non-linear association with SET scores, as students who were 25 years old give the best SET scores, while younger and older students rate their teachers with a lower SET score. Most 25-year-old students are in a master's program and potentially receive more dedicated and specialized teaching that may result in them being more forgiving, resulting in higher SET scores. Section 5.2 discusses the importance of non-linear relationships and interaction in more detail.

**5.1.6 Teacher assessment of students compared to normal.** When teachers were asked to look at student performance compared to normal, we see a clear signal in the SET score model, in the sense that teachers who assess their students as performing much better than normal also receive higher SET scores. It seems natural that students who impress their teachers are also treated in a positive way that also causes them to evaluate their teacher more positively. Interestingly, however, if the teacher assessed the students as performing better than usual, the students did not get higher grades. We have no good explanation for this.

**5.1.7 Administrative data versus survey data.** In our basic models, we include both administrative data and survey data derived from many different questions. We also introduce alternative models, some which only use administrative data. Interestingly, the alternative model for which only administrative data is used does not perform much worse than the full model (Table 3).

However, we see that several interesting and meaningful features of the survey do indeed have a significant impact, such as teacher assessment of the course and students, as well as their own skills and their attitudes toward online teaching. Thus, we find it justified and valuable to add additional survey data of this kind in this type of educational data mining and learning analytics studies.

## 5.2 Implications for learning analytics and educational data mining

**5.2.1 Importance of non-linearity and interaction.** In general, we find it intriguing that features often used as independent variables in learning analytics, such as age of students and age of teachers, have non-linear association with both SET scores and student grades. Although it is not new that non-linearity exists in educational data mining and learning analytics [25, 26], the features that actually interact or have a non-linear association have not been discussed previously, to our knowledge. In this paper, we have tried to bridge this gap by using techniques from interpretable machine learning to investigate interactions in more detail.

Our models show that that being of average age, for both students and teachers, have an apparently positive association with both student grades and SET scores (Sections 4.2 and 5.1.5); that is, age has a non-linear association with the two targets in our models. This aspect is clearly something that should be taken into account when analyzing student learning and teachers performance. Thus, research should incorporate models that can deal with both non-linearity and interaction to a greater extent.

**5.2.2 Measuring performance and quality in education.** The success of the transformation to online teaching during the COVID-19 lockdown can be looked at from at least two

perspectives: 1) How well did the students perform compared to normal? And 2) How well did the teachers perform compared to normal? This study attempted to address both questions.

Grades are used widely in the learning analytics and educational data mining literature ([20]) as a measure of student performance, and we agree overall that it often can be used as such. We also included a question in the teacher survey (Section 3.1.2) on the general assessment of students compared to normal which was aggregated at course level. The SET scores in our study can also be seen as an appropriate way to evaluate teachers' performance in the classroom. However, a measure of performance in relation to the two questions above will, of course, always be proxies. High SET scores and grades might not always reflect the good performance of quality teaching, the quality of a course, or the learning of the students [49].

As the output shows, student features can be associated with teachers' SET scores (though perhaps not their actual teaching quality), and teacher features can be associated with grading. In this regard our findings are interesting since the SET score and grade analyses do not seem to overlap in terms of explaining features, as seen in Section 4. Apparently, what constitutes good teacher performance and good student performance measured using these standard targets are not the same. An understanding of teaching quality needs to take both into account—at least during the COVID-19 lockdown. This approach may be generalizable and should be investigated further. Interestingly, teachers' general assessment of students' performance in class (compared to normal) seems to be the only feature that comes close to being significant for both targets. This could therefore be considered as a potential future measure of teaching quality for teacher surveys.

Other measures that are not included in this study but that can partly serve as proxies for the quality of education are, for example, employment among graduates and their salaries. However, these measures are difficult to compare across educational institutions, and even within the same country or over time, as jobs may be easier to find in some regions than others, just as some sectors may pay more than others. Additionally, we cannot measure them until several years after a study is over.

## 5.3 Limitations

Since the COVID-19 lockdown represented an experimental setting that did not allow randomized controlled trials, we acknowledge that our study has certain limitations and that it does not constitute a full causal understanding of the effect of the lockdown on teaching. First, there are confounding factors that may have had an unwanted effect. As already discussed, it seems that some teachers might have given better marks, for example, out of sympathy due to the extreme situation. The same could apply the other way round with SET scores. Second, by restricting the population to teachers who responded to the teaching survey (and students who did a student evaluation survey), we risk self-selection bias. Although the demographics of teachers and students in data in the models were similar to the rest of the university, some groups were strongly underrepresented, e.g., part-time professors (Table 1). This means that we may have missed out on important insights from this group of teachers, who may have had a different view on teaching because of their circumstances differed from their full-time jobs outside the university sector. Finally, we chose a hypothesis-free approach and included most of the available features that could have had an effect. This tends to lead to spurious relationships, especially in linear models, which means that the estimates should be interpreted with care. However, random forest models are not as sensitive to this, and we see that the two types of models we apply agree in many ways in the analysis of both SET scores and grades.

## 6. Conclusion

This study analyzed student and teacher performance in higher education using the COVID-19 lockdown as a case. We created linear regression models and random forest models, in addition to conducting an interaction analysis on the latter. We did this to address our research question of whether it is possible to combine data sources to identify linear and non-linear features that could improve understanding about performance measured using SET scores and grades during the unique situation of a lockdown at a major Danish business school. We found that the age of both teachers and students generally seemed to have a non-linear association with both targets, besides interacting with other features in the analysis. In addition, we found that linear features such as gender, among both teachers and students, and peer consulting among teachers have a significant association with student grades in particular. The grade model has many significant linear and non-linear features, which reflects that there is a complex underlying mechanism. Assessment of students by teachers and the extra time teachers spent on preparing their courses had a positive association with the SET scores of teachers. The only somewhat significant overlapping feature between the two models using student grades and SET scores as targets is teacher assessments of the students (per course), which means it might be a useful measure for future performance analyses.

## Supporting information

**S1 Fig. Histogram of the grade response variable.**
(PDF)

**S2 Fig. QQ-plot of the grade target variable.**
(PDF)

**S3 Fig. Histogram of the SET target variable.**
(PDF)

**S4 Fig. QQ-plot of the SET response variable.**
(PDF)

**S1 Table. Selected questions for the educators' survey.**
(PDF)

**S2 Table. Demographics of teachers in the grade model.**
(PDF)

**S3 Table. Demographics for teachers in the SET-score model.**
(PDF)

**S4 Table. Background demographics of the students included in the grade models.**
(PDF)

**S5 Table. Background demographics of the students included in the SET-score model.**
(PDF)

**S6 Table. Table of linear regression coefficients, grade model.**
(PDF)

**S7 Table. Table of linear regression coefficients, SET-score model.**
(PDF)

**S1 Text. Elaboration of data pre-processing.**
(PDF)

**S2 Text. Description of alternative model results.**
(PDF)

## Acknowledgments

We would like to thank the Teaching and Learning Unit and the Business Information and Analytics Department at Copenhagen Business School for help with data collection and management. We would also like to thank Annemette Kjærgaard, Mette Frank, Mai Ajspur, Christian Theil Have and Kasper Risager for academic discussions. Finally, we would like to thank the two anonymous reviewers.

Part of the computation done for this project was performed on the UCloud interactive HPC system, which is managed by the eScience Center at the University of Southern Denmark.

## Author Contributions

**Conceptualization:** Sine Zambach, Jens Ulrik Hansen.

**Data curation:** Sine Zambach, Jens Ulrik Hansen.

**Formal analysis:** Sine Zambach, Jens Ulrik Hansen.

**Investigation:** Sine Zambach, Jens Ulrik Hansen.

**Methodology:** Sine Zambach, Jens Ulrik Hansen.

**Project administration:** Sine Zambach.

**Resources:** Sine Zambach, Jens Ulrik Hansen.

**Software:** Sine Zambach, Jens Ulrik Hansen.

**Validation:** Sine Zambach, Jens Ulrik Hansen.

**Visualization:** Sine Zambach, Jens Ulrik Hansen.

**Writing – original draft:** Sine Zambach, Jens Ulrik Hansen.

**Writing – review & editing:** Sine Zambach, Jens Ulrik Hansen.

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
