## [Decision Letter · Decision Letter 0]

20 Jul 2023

PONE-D-23-17797Student and teacher performance during COVID-19 lockdown: An investigation of associated features and complex interactions using multiple data sourcesPLOS ONE

Dear Dr. Zambach,

Thank you for submitting your manuscript to PLOS ONE. After careful consideration, we feel that it has merit but does not fully meet PLOS ONE’s publication criteria as it currently stands. Therefore, we invite you to submit a revised version of the manuscript that addresses the points raised during the review process.

Adjust the paper according to reviewers remarks.

We look forward to receiving your revised manuscript.

Kind regards,

Radoslaw Wolniak, full professor

Academic Editor

PLOS ONE

Journal Requirements:

3. Please include a copy of Table 6 which you refer to in your text on page 18.

Reviewers' comments:

Reviewer's Responses to Questions

**Comments to the Author**

1. Is the manuscript technically sound, and do the data support the conclusions?

Reviewer #1: Partly

Reviewer #2: Yes

2. Has the statistical analysis been performed appropriately and rigorously? 

Reviewer #1: Yes

Reviewer #2: Yes

3. Have the authors made all data underlying the findings in their manuscript fully available?

Reviewer #1: Yes

Reviewer #2: Yes

4. Is the manuscript presented in an intelligible fashion and written in standard English?

Reviewer #1: Yes

Reviewer #2: Yes

5. Review Comments to the Author

Reviewer #1: You have a very good work, deserving publication. Nonetheless, some amendments in the format are due to admit publication. The main problem is the epistemological structure (why the article was conceived and how the study was developed). I suggest the following structure of objectives: (i) research gap; (ii) research question; (iii) purpose of the article; (iv) intermediate objectives; (v) assumptions or hypo; and (vi) research method. This structure must appear in the introduction (you have some parts, but it needs to be put together).

The research gap must be created by a systematic literature review that provides 'holes' in the state of knowledge on the topic. I believe that a full review should not be done, but an analysis of about 5-8 studies on the topic under discussion. You can find some examples, which will show the relevance of the issue, as it is indeed a topic of current, relevant research. At the end of the justification you should write something like: According to what we were able to find, there are no studies referring and reporting on ... With this you have therefore proven that the issue is relevant, and you have also proven that your study does indeed fill a research gap.

I propose the classic division of a scientific article:

Introduction

Literature review

Research methodology

Results

Discussion

Conclusions

Your division into many chapters and subchapters creates chaos in your work.

Reviewer #2: Dear Author

Paper is good (very good - my opinion)

The research topic is a summary of the situation after COVID-19. The research is implemented correctly. As a reviewer I am missing a completed literature review after the Introduction

One note: Figures have a poor resolution

6. PLOS authors have the option to publish the peer review history of their article (what does this mean?). If published, this will include your full peer review and any attached files.

Reviewer #1: No

Reviewer #2: No

---

## [Author Response · Author response to Decision Letter 0]

17 Aug 2023

We have responded to the reviewer to the best of our knowledge in the submission. See the document "Responce to reviewers.docx".

---

## [Editor Report · Decision Letter 1]

4 Sep 2023

Student and teacher performance during COVID-19 lockdown: An investigation of associated features and complex interactions using multiple data sources

PONE-D-23-17797R1

Dear Dr. Zambach,

We’re pleased to inform you that your manuscript has been judged scientifically suitable for publication and will be formally accepted for publication once it meets all outstanding technical requirements.

Kind regards,

Radoslaw Wolniak, full professor

Academic Editor

PLOS ONE
---

## [Editor Report · Acceptance letter]

16 Oct 2023

PONE-D-23-17797R1 

Student and teacher performance during COVID-19 lockdown: An investigation of associated features and complex interactions using multiple data sources 

Dear Dr. Zambach:

I'm pleased to inform you that your manuscript has been deemed suitable for publication in PLOS ONE. Congratulations! Your manuscript is now with our production department. 

Kind regards, 

on behalf of

Professor Radoslaw Wolniak 

Academic Editor

PLOS ONE